# Lignin as a UV Light Blocker—A Review

**DOI:** 10.3390/polym12051134

**Published:** 2020-05-15

**Authors:** Hasan Sadeghifar, Arthur Ragauskas

**Affiliations:** 1R&D Laboratory, Hollingsworth & Vose (H&V) Company, West Groton, MA 01452, USA; 2Center for BioEnergy Innovation (CBI), Oak Ridge National Laboratory (ORNL), Oak Ridge, TN 37831, USA; 3Joint Institute for Biological Sciences, Biosciences Division, Oak Ridge National Laboratory (ORNL), Oak Ridge, TN 37831, USA; 4Department of Chemical and Biomolecular Engineering, The University of Tennessee, Knoxville, TN 37996-2200, USA; 5Department of Forestry, Wildlife and Fisheries, Center for Renewable Carbon, The University of Tennessee Institute of Agriculture, Knoxville, TN 37996-2200, USA

**Keywords:** lignin chemistry, lignin chromophores, sunscreen, UV protection film, sunlight protection

## Abstract

Lignin is the by-product of pulp and paper industries and bio-refining operations. It is available as the leading natural phenolic biopolymer in the market. It has chromophore functional groups and can absorb a broad spectrum of UV light in range of 250–400 nm. Using lignin as a natural ingredient in sunscreen cream, transparent film, paints, varnishes and microorganism protection has been actively investigated. Both in non-modified and modified forms, lignin provides enhancing UV protection of commercial products with less than a 10% blend with other material. In mixtures with other synthetic UV blockers, lignin indicated synergic effects and increased final UV blocking potential in compare with using only synthetic UV blocker or lignin. However, using lignin as a UV blocker is also challenging due to its complex structure, polydispersity in molecular weight, brownish color and some impurities that require more research in order to make it an ideal bio-based UV blocker.

## 1. Introduction

Lignin is a phenolic bio-polymer in the structure of woody plants and most other terrestrial plants which lacks an organized 2° or 3° structure [1,2,3]. However, it is well known that lignin is a polymer of mainly three types of phenyl propanoid structures, including sinapyl alcohol (S type), coniferyl alcohol (G type), and para-coumaryl alcohol (H type) units (Figure 1) [2] linked together primarily by arylglycerol ether linkages and other bonds to form the macromolecule structure. The ratio of these precursors in the lignin structure is dependent on the type of plant resource and extraction protocol used. The exact structure of lignin in the plant is hard to define, and almost all of the proposed lignin structures are model structures based on chemical and spectroscopic analysis of extracted lignin or whole-cell NMR analysis. [1]

Commercially available lignin is primarily a by-product of pulping or to a lesser extent bio-refinery process such as cellulosic ethanol [1]. Depending on the pulping process, two categories of the lignin are available, including alkali and sulfonated lignin. Alkali lignin including soda and kraft lignin are not soluble in water, but lignosulfonate dissolves easily due to the presence of hydrophilic sulfonate groups. The biorefining process also can yield either an extracted lignin or a residual lignin after pretreatment, biological deconstruction and conversion of cellulosic material [4]. During pulping or the biorefining process, the structure of lignin in the plant can undergo degradation forming smaller molecules with new functional groups. In this processing chemistry a variety of new chromophores are introduced into lignin, especially during harsh pulping conditions, which are important factors that are responsible for the dark color of chemical pulping lignin [5,6].

Technical lignin from the pulping process contain UV chromophore functional groups including quinones and methoxy substituted phenoxy groups, that can be conjugated with double bonds or carbonyl functional groups [3,7,8,9]. Unsaturated functional groups that absorb visible light are the main light absorbers and make lignin’s color brownish to black. These groups include conjugated carbonyl groups, aromatic rings and carbon–carbon double bonds [10,11]. A spectrum of lignin light absorption in the UV and visible light range and different chromophore groups with their approximate wavelength absorptions are indicated in Figure 2 [6,10,11]. 

The presence of lignin turns ligno-cellulosic fibers to a darker color (i.e., photo-yellowing) after exposure to photo-irradiation [12]. Although this property is a negative effect for wood-based pulps (i.e., mechanical pulps) but it can be a positive effect for lignin application as a UV blocker. One of the main mechanisms for photo yellowing of lignin is light-catalyzed photo-reactions with lignin chromophores ultimately resulting in the formation of quinones and other chromophoric bodies [12,13]. This phenomenon can accelerate lignin UV blocking potential even after exposure to sunlight. However, the dark color of lignin is a challenge for its application and can reduce final product transparency. To minimize the lignin color issue, different solutions have been proposed including lignin fractionation to collect less discolored fractions of lignin [14] and modification of lignin through acetylation, which can reduce lignin color without effecting its UV light absorption properties [15].

Radiation of UVA (320−400 nm) and UVB (280−320 nm) from sunlight increases biological damage of skin, degradation of organic compounds, discoloration of dyes and pigments, weathering and yellowing of plastics and films, loss of mechanical properties (i.e., cracking) and other problems associated with UV irradiation [7,16,17,18]. Manufacturers are interested in offering products that remain unaltered for long periods under severe sunlight exposure conditions [19,20]. UV absorbers are necessary to reduce these damaging photo effects. The active ingredients in a wide range of these applications are often synthetic organic chemicals and minerals. Due to the unexpected side effects of such chemicals, the search for greener alternatives has been receiving significant attention recently. Natural products, such as green coffee oil and extracts of *Carica papaya* and *Rosa kordei*, have been indicated as useful sunlight radiation protection functions [21,22]. UV protection is usually calculated by the sun protection factor (SPF), which is defined in Equation (1):(1)$$\mathrm{SPF}=\overset{400}{\underset{290}{\sum }}E_{\lambda }S_{\lambda }/\overset{400}{\underset{290}{\sum }}E_{\lambda }S_{\lambda }T_{\lambda }$$
Where, *E_λ_ = CIE* erythemal spectral effectiveness, S_λ_ = solar spectral irradiance, and T_λ_ = spectral transmittance of the sample [7]. The percentage of UV protection is equal to 100 − (100/SPF). For example, a material with SPF = 15 means it can protect 93.33% of UVB light (100 − (100/15)). 

Using lignin as a UV blocker in the sunscreen has been frequently reported. The addition of lignin could increase the SPF of sun lotions with broad-spectrum UV-absorbing and antioxidant properties. However, lignin derived from the pulping process is a dark heterogeneous material with minerals and organic impurities which requires lignin modification, purification, and/or fractionation to improve its UV blocking performance and application [3,8,9,23].

Sun blocker creams in the market frequently contain minerals like zinc oxide (ZnO_2_) and titanium dioxide (TiO_2_) or a wide range of organic chemicals to block UV radiation [24]. Due to the adverse effects of synthetic UV blockers on skin tissue, using a natural blocker has seen increasing attention recently [8]. Many natural polyphenol extracts have been investigated as a UV blocker. However, most of the natural products have small molecules with low purity, have poor photostability [7], are partial spectrum sun blockers and cannot block the full spectrum of UV light. Therefore, more stable natural macromolecular sunscreens are desired.

To protect materials from solar radiation, different commercial organic and inorganic packaging film materials have been developed with UV protection. The primary substrate of the film or packaging materials are synthetic polymers. Developing a greener generation of natural-based self-protective transparent UV filters is essential due to the disadvantages of petroleum-based polymers [25]. Cellulose-based films containing UV absorbers have been reported to have potential applications as outdoor UV-sensitive polymers, car windshields, clean windows, contact lenses, special biological test containers and as an alternative for synthetic polymers. The UV protection of paint and varnish with lignin is reported as a greener alternative for synthetic UV blockers. Finally, using lignin as a UV shield to protect microorganisms has been reported.

## 2. Lignin as a Component in Sunscreen Product

Evaluation of softwood (SW) alkaline lignin after washing and purification with pure water and acidic solution was reported [26] for the development of high-performance broad-spectrum sunscreen. Lignin was added to several commercial sunscreen products and significant enhancements in ultraviolet (UV) absorbance were reported. Adding 2 wt% lignin to a SPF 15 sunscreen doubled its SPF to 30. Adding 10 wt% Kraft lignin increased its SPF to 50. Enhanced sunscreen performance after radiation with UV light was also reported. After 2 h of UV radiation, UV absorbance of 10 wt% lignin-SPF 15 sunscreen increased by more than 40%. This result has been attributed to specific synergistic effects between lignin and other ingredients in sunscreen lotions, as well as the antioxidant property of lignin.

Industrial lignin (i.e., alkali lignin and lignin with low sulfonate content) were blended with a pure hand cream and a commercial sunscreen lotion [27]. Lignin was found to significantly boost their sunscreen performance. The hydrophobic nature of alkali lignin indicated better sunscreen performance than hydrophilic counterpart (sulfonated one). Sun protection factor (SPF) of the pure hand cream (SPF 1.1) containing 10 wt% sulfonated lignin reaches to 8.66 (around 90% UV light blocking). In a mixture with sunscreen lotion, even the addition of 1% lignin almost doubles its SPF. Studies have also found that hydrophilic lignin tends to demulsify the lotions due to an electrostatic disequilibrium. After 2 h of UV radiation, UV absorbance of lignin-modified sunscreen lotions were increased up to 40%. Lignin was found to have a general synergistic effect with sunscreen actives in commercial lotions, and increased UV blocking potential even after irradiation with UV.

In another effort to use lignin micro particles from organic acid lignin as a UV absorber [28], a significant SPF value increase in pure hand lotion from SPF 1 to 3.53 at 5 wt% dosage of lignin was reported. Furthermore, the UVA/UVB ratio (0.69–0.72) for this lignin indicated that they exhibit superior properties. It was claimed that lignin particles, with excellent antioxidant and UV protection capacities, are a natural source for sunblock cosmetics.

In another study, lignosulfonate (LS) was applied to modify TiO_2_ as a mineral sunscreen [29]. The results showed that esterification occurred between the carboxyl groups of lignin and the hydroxyl groups on the surface of TiO_2_. The coated lignin on the TiO_2_ surface not only improved the dispensability of TiO_2_ in the substrates but also significantly boosted its UV-blocking ability. TiO_2_ coated with lignin nanocomposites was applied in a pure hand cream, and the sunscreen performance was studied against TiO_2_ as control. The sun protection factor (SPF) values of creams containing 5 wt%, 10 wt%, and 20 wt% lignin on TiO_2_ were 16, 26, and 48, respectively.

Composites of alkali lignin (from a commercial source) and kraft lignin (from agri-biomass) with different zinc compounds (i.e., zinc acetate, zinc sulphate hexahydrate, zinc nitrate heptahydrate, zinc oxide nanoparticles (ZnONP)) were reported as UV blockers in a mixture with a hand cream [30]. A 20% mixture of just lignin with cream showed around 7% transmittance in the UV range of 200–350 nm (93% blocking) whereas commercial ZnONPs showed 10–25% transmittance in the entire UV range (75–90% blocking). Using both lignin and zinc products nanocomposites showed 0% transmittance in the UV range of 200–350 nm and 5%–15% transmittance between 350–400 nm. They concluded that the synergistic effect of lignin and zinc oxide played a key role in imparting excellent UV blocking potential to the lignin derived zinc oxide nanocomposites.

A sunscreen cream prepared with bagasse soda lignin and ZnO nanoparticles in a mixture with pure hand cream is reported as a good sunscreen cream [31]. They formulated different creams with 10% ZnO nanoparticles and 5%, 10% and 15% lignin. Pure cream SPF was around 1.1 (10% absorbance). The addition of 15% lignin increased its absorbance up to 88% (SPF 8), which is the same as using 15% ZnO. However, mixing 5 wt% of ZnO and 15 wt% of the lignin together increased UV absorbance to 92% (SPF of 12.5).

Kraft lignin-based sunscreen creams were prepared by blending 10% lignin (or modified lignin colloidal particle) with 90% pure Nivea cream at 300 rpm for 20 h at room temperature in darkness [32]. The critical wavelength (*λc*) of the optimal sample was ∼375 nm, while the UVA/UVB ratio reached 0.84. By using 130 nm colloidal lignin particles, SPF of the cream was increased to 56. Superior photostability was also reported. In the same study, applying lignin nanoparticles from rice husk into moisturizing cream (Nivea) at concentrations of 1 and 5 wt% is reported to make a lignin-based sunscreen [33]. At 5 wt% lignin nanoparticles, the cream exhibited SPF values about twice as high as those of the pure cream. When lignin nanoparticles were added into commercial sunscreen (5 wt%), the UV protection factor increased by a factor of five overall (from SPF 5.4 to SPF 30.0).

An enzymatic lignin product modified with ozone oxidation grafted with polyethylene glycol was reported to exhibit excellent antioxidant and UV absorbing properties in the UVA and UVB spectral region [34]. The ozone oxidation process provided an effective and green method for the preparation of highly reactive lignin, which can be applied in the manufacturing of a multifunctional lignin polymer.

## 3. Lignin as UV Blocker for Packaging and Transparent Films

Preparation of a lignin-containing flexible and transparent film using gellan gum, 2-hydroxyethyl cellulose (HEC), and softwood kraft lignin composite has been reported for food packaging and biomedical applications [35]. Incorporating 10 wt% lignin in the composite film provided high ultraviolet (UV) blocking properties, with almost 100% protection against UVB (280–320 nm) and 90% against UVA (320–400 nm). In addition to UV protection, the presence of the lignin improved the thermal, mechanical, and hydrophobic properties of the film. They also reported excellent radical scavenging and non-cytotoxic activities for the film containing 1 wt%, 5 wt%, and 10 wt% lignin.

Alkali lignin with low sulfonate content/poly(vinyl alcohol) (PVA) electro spun fibers with different lignin concentrations were reported to have antimicrobial and ultraviolet (UV) absorption properties [36]. The lignin was crosslinked with PVA so as to be stable in an aqueous media. The composite mixture exhibited ultraviolet protection factors (UPF) of >50, indicating excellent UV protection.

Poly(methyl methacrylate) film reinforced with coconut shell lignin fractions (i.e., acetone and ethanol soluble) was reported for its UV-blocking properties [37]. The incorporation of lignin increased the UV absorption capacity of the prepared film drastically. The film containing lignin showed the around 26% lower transparency than pure poly(methyl methacrylate) films, but transmitting about 79% of the radiation at 550 nm, and exhibited excellent UV absorption capability. A film prepared with 1% ethanol soluble fraction of lignin indicated 40% transparency (60% UV-blocking capacity) at 400 nm wavelength. The authors proposed that the discoloration of lignin can be attributed to two main reactions. First, the phenolic hydroxyl groups in lignin under the UV exposure generate phenoxyl radicals, which will be oxidatively converted, in part, to quinones. Posteriorly, as quinones are efficient chromophores, their photo-oxidization into aliphatic acid structures will lead to the bleaching of lignin [9]. The UV-absorption capacity of the films remained practically unchanged in the UVB range and had a slight reduction of UV absorption in the UVA range.

Preparation and characterization of semitransparent flexible cellulose films bonded covalently with lignin indicated high UV light blocking potential [8]. Azide modified cellulose was dissolved in dimethylacetamide/lithium chloride (DMAc/LiCl) and reacted with propargylated lignin to produce 0.5 wt%, 1 wt%, and 2 wt% lignin-containing materials. Cellulose-lignin films were prepared by regeneration in acetone to make a homogeneous film structure. Prepared films indicated a high UV protection ability. Cellulose film containing 2 wt% lignin showed 100% protection of UVB (280−320 nm) and more than 90% of UVA (320−400 nm). The UV protection of prepared films was persistent when exposed to thermal treatment at 120 °C and UV irradiation for 2 h.

A straightforward procedure to produce renewable-based cellulose-lignin UV-light-blocking films through the mixing of tannin and lignin with polypropylene (PP) using a dynamic vulcanization technique. The resulting films were examined for mechanical and UV blocking properties [31]. The results indicated dynamic vulcanization as a powerful tool to make a lignin-PP blend with more compatibility. Furthermore, vulcanized tannin/lignin present better UV shielding performance with fewer changes of surface morphology, carbonyl index, crystallinity, viscosity, and tensile property. The surface of the composite containing lignin and tannin was exposed at 0, 168, and 336 h UV light in comparison with pure PP which indicated that the composites had less degradation against UV irradiation over the PP control.

A film made from a composite of lignosulfonate and clay minerals without requiring a petroleum-based component has been reported [38]. A thin film was obtained through a simple one-step procedure involving knife-casting of lignin mixed with water-dispersed natural clay minerals. The produced film exhibits excellent UV protection properties (i.e., 99% UVA protection).

As an alternative approach, laccase-catalyzed oxidative polymerization of dimeric lignin model compounds yielding products with a number-average molecular weight (*M*_n_) 700 to 2300 Da was investigated as UV blockers in mixtures with polyvinyl chloride (PVC) as a thin film [39]. The oligomers provided good UV light absorption characteristics with a high molar extinction coefficient (5000–38000 m^−1^cm^−1^) in the UV spectral region, especially above 300 nm. Additionally, the product showed good photo stability in accelerated UV weathering experiments.

Production of biodegradable and UV blocker transparent film was reported using alkaline lignin (AL) and softwood kraft lignin (SKL) and nano crystalline cellulose (CNC) at various concentrations (1−10 wt%) [14]. Product analysis by SEM indicated homogeneity of the lignin distribution in CNC/lignin films. Using 10% lignin in the film provided complete UV blocking in UVA spectra up to 400 nm. The researchers also reported results for acetylated lignin. Using acetylated lignin reduced the color of lignin without significantly affecting the UV-absorption properties. The presence of lignin also improved the thermal stability of the films. The maximum degradation temperature of the lignin containing sample increased from 300 to 330 °C.

Choline citrate, a bio-based ionic liquid, (IL) stabilized homogeneous gelatin–lignin UV shielding films, with excellent antimicrobial and mechanical properties reported [40]. The antimicrobial activity of the prepared films was tested against Bacillus subtilis. The prepared biofilms indicated a sun protection factor (SPF) of up to 45. The addition of 0.5 wt% to 1.5 wt% lignin in a gelatin matrix increases the SPF from 7 in pure gelatin to 39.0–44.6, which the researchers compared to other synthetic polymer materials.

A biodegradable film derived from the grafting of soda lignin with 10-undecenoic and oleic acids were reported via solvent and catalyst-free processes [41]. The resulting lignin ester derivatives and raw lignin were melt-blended with a biodegradable poly(butylene adipate-*co*-terephthalate) (PBAT) to prepare UV protective films. The new composite containing lignin exhibited good dispersion of lignin particles with excellent mechanical properties. The prepared films indicated a UV-barrier property with 10 wt% lignin loading in a wide range of UV light (280–400 nm). The UV protection of prepared films proved persistent even after UV irradiation for 50 h, and their transparency was enhanced.

TiO_2_ particles decorated with lignin (TiO_2_@lignin) were synthesized by the hydrothermal method in aqueous solution to improve the UV shielding performance of lignin particles [42]. The poly(propylene carbonate) (PPC) composite films (thickness of ~23 μm) with different contents of TiO_2_@lignin were prepared via a blade-casting method. Uniform dispersion of TiO_2_@lignin in the PPC matrix with good miscibility was reported. UV–vis transmission spectra results revealed that the PPC composite film containing 5 wt% TiO_2_@lignin absorbed about 90% of UV light in the broad UV spectrum (200–400 nm), indicating the TiO_2_@lignin had a good UV-shielding property.

A composite film from a sonicated mixture of lignin nanoparticles (LNP) with polyvinyl alcohol (PVA) aqueous solution was reported to yield films with lignin content from 0.5 to 10 wt% [43]. The UV blocking test results indicated high UV absorption with higher than 97.5% UV blocking. These results demonstrated the prospective of using lignin-containing films as attractive biomass-based packaging and coating material, having excellent transparency and UV protection capability, mostly essential for light-sensitive products

The preparation of biomass-based films through reactive compatibilization of lignin with poly (lactic acid) (PLA) has also been examined [44]. Using a facile and practical route, the hydrophilic hydroxyl groups of lignin were acetylated to impose the required compatibility with PLA. The solubility parameter of the pristine lignin at 26.3 (J/cm^3^)^0.5^ in chloroform (CF), was altered to 20.9 (J/cm^3^)^0.5^ by acetylation, allowing good compatibility with PLA at 20.2 (J/cm^3^)^0.5^. The improved compatibility of lignin and PLA provided substantially decreased lignin domain size in composites (12.7 μm), which subsequently gave transparent UV-protection films (visual transmittance at 76% and UV protection factor over 40). The developed PLA/lignin composites provided a 100% biomass derived film with balanced optical and mechanical properties which could broaden its eco-friendly applications in various applications.

## 4. Lignin as UV Blocker for Varnish and Oil Coated Materials

A UV blocker for varnish is reported using pine wood organosolv lignin (OL) prepared as nano-colloidal particles [45]. The UV transmittance of the varnish with 1 wt% OL was lower than that of a commercial varnish containing synthetic UVA and UVB blocker. However, UV irradiation of a 10 wt% lignin-containing sample for 100 h showed 86% for UVA and 69% for UVB with less color change, compared to that of pure and commercial varnishes. A colloidal formulation of lignin indicated better compatibility and UV-blocking and photostability than the original raw lignin. Additionally, 10 wt% of lignin increased the hardness and modulus of elasticity of varnish by 61% and 36%, respectively. Modified enzymatic hydrolysis lignin through sulfomethylation in combination with alkyl polyglucoside (APG) was reported as an emulsifier to stabilize the oil-in-water (O/W) and UV protection [46]. The product containing lignin exhibited superior UV protection of at least 30% higher than the control sample after 72 h of UV irradiation.

## 5. Lignin as Microorganism Protection

The role of lignin on UV protection of microorganisms and enhancing their growth has also been reported [47,48,49]. The vital role of lignin in protecting *Escherichia coli* upon UV induced mortality has been demonstrated [49]. They reported that in the absence of lignin, *E. coli* indicated 100% mortality when irradiated with UV light for 5 min. However, in the presence of 5 wt% lignin nanoparticles, the survival rate of *E. coli* was 97% after 5 min of UV irradiation. They also reported better results when they used lignin nanoparticles instead of regular macro-sized lignin particles. The same work has reported protecting specific bacteria, fungi, and viruses as pest control [50]. These microbes are rapidly degraded by sunlight, which limits their field efficacy. Adding lignin into the pesticide formulations indicated high protection against sunlight energy, specifically the ultraviolet wavelengths. Protecting of entomopathogenic fungus, *Beauveria bassiana*, which is highly susceptible to solar radiation by lignin has been reported [51]. Fungi spores were coated with lignin by spray release. Rates of loss in spore viability under simulated solar radiation were approximately ten times lower for the non-coated spore. Effect of lignin on the protection of *Pichia anomala* strain K and antagonistic yeasts as biocontrol agents of postharvest fruit diseases against UV radiation was investigated [52]. UVB radiation (280 to 320 nm) can significantly reduce yeast survival and effectiveness. Lignin reduced yeast mortality caused by UVB radiation on apple fruit surfaces.

## 6. Conclusions

Lignin is a by-product of biorefining and the chemical pulping industry and this material provides good potential as a natural UV protection ingredient in broad-spectrum (UVA, UVB) sunscreens. Mixing lignin with lotion and cream indicated an excellent range of sun protection factor (SPF). As a sunscreen ingredient, Lignin indicated more than 95% UV protection with adding 5−10 wt% in pure white hand creams. Especially when lignin was mixed with other commercial sunscreens in the market, in a synergistic effect with other ingredients, a low dosage of lignin (1−3 wt%) dramatically increased SFP. Lignin composites with synthetic polymers or natural polymers like cellulose and starch indicated good potential for transparent UV blocking films, with the addition of less than 10% lignin in the composite. Small molecular weight fraction of lignin fractionatio and lignin in form of micro- and nano-particles, indicated better UV blocking potential than non-processed lignin. Lignin indicated a high potential to protect paint, oil, and varnish from UV degradation and protect microorganism’s growth against sunlight.

## Figures and Tables

**Figure 1 polymers-12-01134-f001:**
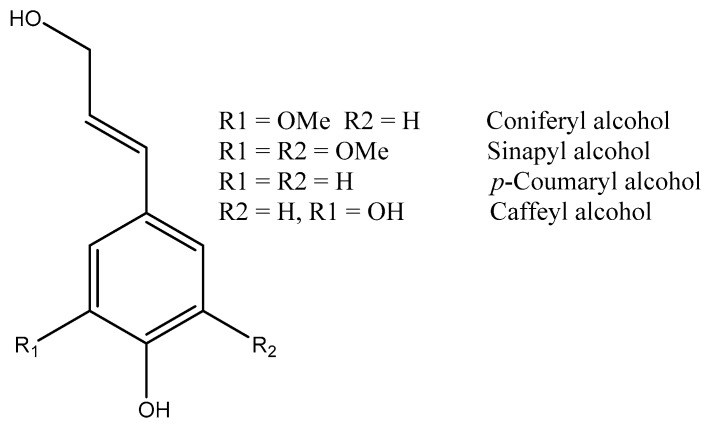
The three building blocks of lignin phenylpropane units structure. Redrawn by the authors [2].

**Figure 2 polymers-12-01134-f002:**
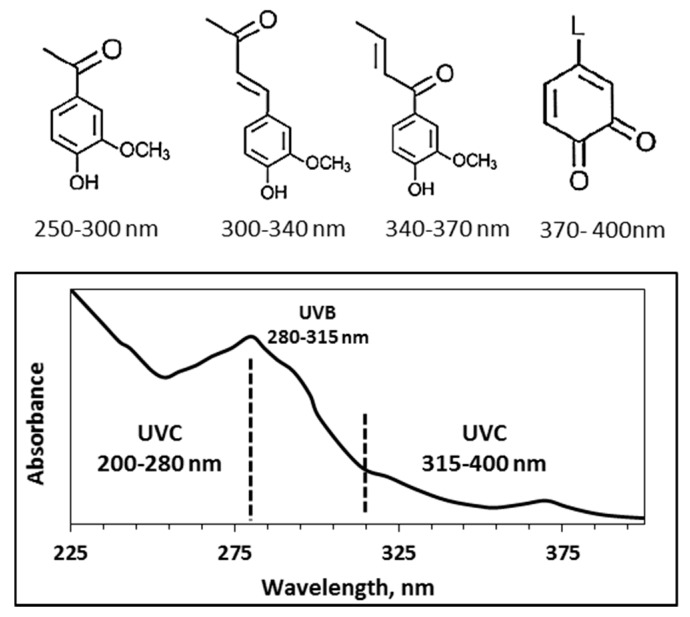
Chromophores in lignin structure and their UV absorption spectra, reprinted and redrawn by the authors [6,10,11].

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
