# Peer review of "Lignin as a UV Light Blocker—A Review"

_polymers, 2020, doi:10.3390/polym12051134_

Round 1

Reviewer 1 Report

I reviewed article entitled  "Lignin as a UV light blocker" by Sadeghifar and Ragauskas. This work reviewed basic background on lignin and some of the previous studies carried out in using lignin as UV light blocker. Overall it looks fine and I guess it can be published as it is.

Author Response

Reviewer 1

I reviewed article entitled “Lignin as a UV light blocker" by Sadeghifar and Ragauskas. This work reviewed basic background on lignin and some of the previous studies carried out in using lignin as UV light blocker. Overall it looks fine and I guess it can be published as it is.

Our response

We appreciate reviewer for taking the time to review the article.

Reviewer 2 Report

Reviewers' comments:

Full Title: Lignin as a UV light Blocker – A review

Comments: 
The manuscript reported on Lignin as a UV light Blocker - A review. The manuscript needs a detailed editing. The authors need to provide answers to the issues listed below before the manuscript should be accepted for publication.

1) In the Abstract: the authors need to improve.

2) Introduction part should be detailed, it is useful for new readers.

3) Figure 1 and 2, is not clear.

4) Check line number: 79 and 80, equation is missing. 

5) Please provides the references for equations and formula.

6) Conclusion should be specific short results.

7) Several faults: are added or missing spaces between words: see manuscript file.

8) References: make all references in same format for volume number, page number and journal name, because it is difficult to searching and reading (for example references, 1) Ragauskas AJ, Beckham GT, Biddy MJ, et al. Lignin valorization: Improving lignin processing in the biorefinery. Science 2014, 80, 344, (6185), 11) Polcin, J. and Rapson, W.H.; Effects of Bleaching Agents on the Absorption Spectra of Lignin in Groundwood Pulps Part 2. Oxidative-Reductive Bleaching. Pulp and Paper Magazine of Canada, 1971, 72 3 80-91, and 38) Wu Y, Qian Y, Lou H, Yang D, Qiu X. Enhancing the Broad-Spectrum Adsorption of Lignin through  Methoxyl Activation, Grafting Modification, and Reverse Self- Assembly. ACS Sustainable Chem. Eng. 2019, 7:15966−15973.

So that I recommended this manuscript to major revision and for future process.

Author Response

Reviewer 2

First of all, we appreciate reviewer for all the great notes and recommends

  1. In the Abstract: the authors need to improve.

Our answer

We put more detail in abstract

2.Introduction part should be detailed, it is useful for new readers.

Generally we do agree with reviewer, however considering the size of review, we think we provided enough data in the introduction section for better understanding of the review, lignin structure and its photochemistry

3.Figure 1 and 2, is not clear

Both figures are redraw and the quality of figures are improved

4) Check line number: 79 and 80, equation is missing. 

Equation is in right place now

5) Please provides the references for equations and formula.

Ref. is added for equation

6) Conclusion should be specific short results.

Conclusion is modified with more details

7) Several faults: are added or missing spaces between words: see manuscript file. Article reviewed again for any error and corrected.

8) References: make all references in same format for volume number, page number and journal name, because it is difficult to searching and reading (for example references, 1) Ragauskas AJ, Beckham GT, Biddy MJ, et al. Lignin valorization: Improving lignin processing in the biorefinery. Science 2014, 80, 344, (6185), 11) Polcin, J. and Rapson, W.H.; Effects of Bleaching Agents on the Absorption Spectra of Lignin in Groundwood Pulps Part 2. Oxidative-Reductive Bleaching. Pulp and Paper Magazine of Canada, 1971, 72 3 80-91, and 38) Wu Y, Qian Y, Lou H, Yang D, Qiu X. Enhancing the Broad-Spectrum Adsorption of Lignin through  Methoxyl Activation, Grafting Modification, and Reverse Self- Assembly. ACS Sustainable Chem. Eng. 2019, 7:15966−15973.

All ref. were reviewed again and all errors were fixed

Reviewer 3 Report

It is a comprehensive summary about the application of lignin as a UV light blocker. The topic is interesting. However, this paper is more like a book chapter for teaching rather than a research paper. Therefore, a major revision is required. I have the following comments for the authors:

  1. More in-depth and insightful discussion is required. The authors should not only present the research status, but also give more comments on what is going on and what is the future development?
  2. I think more references should be included. 59 is not enough for a review paper. Besides, please have more recent publications.
  3. Being a researcher in the filed of civil engineering, I’d like to see some reviews about the anti-UV application of lignin in infrastructures. Please kindly insert some papers about such application.

Author Response

Reviewer 3

First of all, we appreciate reviewer for all the great notes and recommends

  1. More in-depth and insightful discussion is required. The authors should not only present the research status, but also give more comments on what is going on and what is the future development?

In agreement with reviewer, it will be good idea to discuss more details and look about the future development, however, this is a review and we only presenting current published articles. We tried to discuss details of all reports as much as possible and also mentioned repeatedly about challenging of using lignin as UV blocker and more future work needed to developing its potential as UV blocker.

  1. I think more references should be included. 59 is not enough for a review paper. Besides, please have more recent publications.

We tried to collect almost all the published articles in the scope of this review and we think we almost covered all published articles directly related to the review subject. We believe that despite limited published article in this field, this review will bring new attention to researcher for more working on this topic and create high citation for the review and “polymers” journal.

  1. Being a researcher in the filed of civil engineering, I’d like to see some reviews about the anti-UV application of lignin in infrastructures. Please kindly insert some papers about such application

We didn’t find articles that directly worked on effect of lignin in infrastructure UV protection application. A good number of work are reported about application of lignin as an additives and emulsifier in asphalt and concrete. We have a plan to write a new review in this field and publish in “polymers” journal soon.        

Round 2

Reviewer 2 Report

The authors revised the manuscript according to the reviewers' comments.

Reviewer 3 Report

I am satisfied with this version.